# Trends in Molecular Diagnosis and Diversity Studies for Phytosanitary Regulated *Xanthomonas*

**DOI:** 10.3390/microorganisms9040862

**Published:** 2021-04-16

**Authors:** Vittoria Catara, Jaime Cubero, Joël F. Pothier, Eran Bosis, Claude Bragard, Edyta Đermić, Maria C. Holeva, Marie-Agnès Jacques, Francoise Petter, Olivier Pruvost, Isabelle Robène, David J. Studholme, Fernando Tavares, Joana G. Vicente, Ralf Koebnik, Joana Costa

**Affiliations:** 1Department of Agriculture, Food and Environment, University of Catania, 95125 Catania, Italy; 2National Institute for Agricultural and Food Research and Technology (INIA), 28002 Madrid, Spain; cubero@inia.es; 3Environmental Genomics and Systems Biology Research Group, Institute for Natural Resource Sciences, Zurich University of Applied Sciences (ZHAW), 8820 Wädenswil, Switzerland; joel.pothier@zhaw.ch; 4Department of Biotechnology Engineering, ORT Braude College of Engineering, Karmiel 2161002, Israel; bosis@braude.ac.il; 5UCLouvain, Earth & Life Institute, Applied Microbiology, 1348 Louvain-la-Neuve, Belgium; claude.bragard@uclouvain.be; 6Department of Plant Pathology, Faculty of Agriculture, University of Zagreb, 10000 Zagreb, Croatia; edermic@agr.hr; 7Benaki Phytopathological Institute, Scientific Directorate of Phytopathology, Laboratory of Bacteriology, GR-14561 Kifissia, Greece; m.holeva@bpi.gr; 8IRHS, INRA, AGROCAMPUS-Ouest, Univ Angers, SFR 4207 QUASAV, 49071 Beaucouzé, France; marie-agnes.jacques@inrae.fr; 9European and Mediterranean Plant Protection Organization (EPPO/OEPP), 75011 Paris, France; petter@eppo.int; 10CIRAD, UMR PVBMT, F-97410 Saint Pierre, La Réunion, France; olivier.pruvost@cirad.fr (O.P.); isabelle.robene@cirad.fr (I.R.); 11Biosciences, University of Exeter, Exeter EX4 4QD, UK; d.j.studholme@exeter.ac.uk; 12CIBIO—Centro de Investigação em Biodiversidade e Recursos Genéticos, InBIO-Laboratório Associado, Universidade do Porto, 4485-661 Vairão, Portugal; ftavares@cibio.up.pt or; 13FCUP-Faculdade de Ciências, Departamento de Biologia, Universidade do Porto, Rua do Campo Alegre, 4169-007 Porto, Portugal; 14Fera Science Ltd., Sand Hutton, York YO41 1LZ, UK; joana.vicente@fera.co.uk; 15Plant Health Institute of Montpellier (PHIM), Univ Montpellier, Cirad, INRAe, Institut Agro, IRD, 34398 Montpellier, France; koebnik@gmx.de; 16Centre for Functional Ecology-Science for People & the Planet, Department of Life Sciences, University of Coimbra, 300-456 Coimbra, Portugal; 17Laboratory for Phytopathology, Instituto Pedro Nunes, 3030-199 Coimbra, Portugal

**Keywords:** *Xanthomonas*, molecular methods, quarantine pests, regulated non-quarantine pests

## Abstract

Bacteria in the genus *Xanthomonas* infect a wide range of crops and wild plants, with most species responsible for plant diseases that have a global economic and environmental impact on the seed, plant, and food trade. Infections by *Xanthomonas* spp. cause a wide variety of non-specific symptoms, making their identification difficult. The coexistence of phylogenetically close strains, but drastically different in their phenotype, poses an added challenge to diagnosis. Data on future climate change scenarios predict an increase in the severity of epidemics and a geographical expansion of pathogens, increasing pressure on plant health services. In this context, the effectiveness of integrated disease management strategies strongly depends on the availability of rapid, sensitive, and specific diagnostic methods. The accumulation of genomic information in recent years has facilitated the identification of new DNA markers, a cornerstone for the development of more sensitive and specific methods. Nevertheless, the challenges that the taxonomic complexity of this genus represents in terms of diagnosis together with the fact that within the same bacterial species, groups of strains may interact with distinct host species demonstrate that there is still a long way to go. In this review, we describe and discuss the current molecular-based methods for the diagnosis and detection of regulated *Xanthomonas*, taxonomic and diversity studies in *Xanthomonas* and genomic approaches for molecular diagnosis.

## 1. Introduction

The genus *Xanthomonas* was created by Dowson in 1939 to gather Gram-negative rods, forming yellow colonies, that are motile by the means of a single polar flagellum. Later on, it was observed that a few *Xanthomonas* lineages form white colonies (e.g., *X. citri* pv. *mangiferaeindicae* and *X. populi*) or produce diffusible pigments, such as the fuscous strains of the bean pathogen, *X. citri* pv. *fuscans*. This genus groups species with the collective ability to infect a panoply of crops and wild plants. The genus *Xanthomonas* belongs to the *Lysobacteraceae* family (syn: *Xanthomonadaceae*) [1], including the genus *Xylella*, which contains the well-known plant-pathogenic species *X*. *fastidiosa* with an extremely wide host range [2], and the genera *Stenotrophomonas* and *Lysobacter* with some species recognized as biological control agents for plant diseases [3,4]. Currently, 31 species are validly described for the *Xanthomonas* genus [5] (https://lpsn.dsmz.de/genus/Xanthomonas accessed on 2 March 2021). Most of the species are of phytosanitary concern since they are causal agents of diseases with a global impact on seed, plant and food trade, being responsible for important economic and environmental losses [6,7]. The contours of this genus have tremendously evolved with the transfer of species into related genera (e.g., [8,9]), the description of novel species [10,11] and taxonomic rearrangements leading to synonymy of taxa [12,13,14,15].

Historically, xanthomonads were described as pathogens, collectively affecting a large diversity of plant species. Nearly 30 years ago, Hayward listed 124 monocots and 268 dicots as hosts of the various described *Xanthomonas* species as described by Leyns [16]. However, the ability of individual strains to cause disease is limited due to their narrow host ranges and/or tissue specificities. This useful knowledge concerning the host range of plant-pathogenic bacteria is translated into the infrasubspecific subdivision called the ‘pathovar’, which groups strains that cause the same disease on the same host range [17]. The pathovar ranking has no taxonomical standing but it is practical for plant pathologists and regulation purposes [18]. At least 125 pathovars have been described within the various species of this genus [19].

*Xanthomonas* spp. are increasingly considered as plant-associated bacteria rather than exclusively as pathogens of plants. Indeed, some *Xanthomonas* strains were isolated from asymptomatic plant material and no symptoms developed following their artificial inoculation on their host of isolation or other candidate hosts [10,11,20,21,22]. These non-pathogenic strains, also sometimes referred to as look-alikes of *Xanthomonas* pathogens, have been isolated from a wide diversity of organs and plant species, such as seeds, buds, leaves from legumes, cereals, and fruit trees [23,24,25,26,27]. Non-pathogenic strains are interspersed with pathogenic ones at an infraspecific level, especially within the *X*. *arboricola* and *X*. *euroxanthea* species [11,28,29,30], which implies that highly specific detection tools are required to avoid confusion that could have harmful socio-economic consequences. The coexistence of pathogenic and non-pathogenic *Xanthomonas* strains in the same host plant largely increased the estimated diversity in the genus [6,19,24,30,31,32], with events such as recombination and horizontal gene transfer contributing to the pathogen diversification across the different pathosystems [28,33,34,35,36,37].

Outbreaks of diseases caused by xanthomonads have been reported for multiple hosts such as bananas, beans, cabbage, cassava, citrus, pepper, rice, tomato, and wheat, responsible for high production losses and threatening the livelihood of millions of farmers [7]. Present, emerging, or re-emerging plant diseases due to *Xanthomonas* infection are continually challenging food security and causing significant losses to the economy every year. Given future climate change scenarios that predict an increase in epidemic severity and a geographical expansion of pathogens, the pressure on agri-food systems will become even more relevant. *Xanthomonas* spp. cause a large range of symptoms that in some cases are not easily distinguishable from those caused by other pathogenic bacteria on the same host plants and include water-soaked spots evolving into necrosis on leaves, wilting, rotting, hypertrophy, hyperplasia, blights, dieback, and cankers [38]. In this context, the development and standardization of detection and diagnostic methods, as well as the knowledge of their diversity, are of utmost importance to fully understand the multidimensional nature of *Xanthomonas* and to implement effective containment and control measures. In this review, we will focus on molecular methods for diagnosis, detection, and studies on the diversity of plant pathogenic *Xanthomonas*, concentrating especially on regulated pathogens in the European Union (Table 1). This work is the collective effort of the “Working Group 1: Diagnostics & Diversity–Population Structure” from ‘EuroXanth’ COST Action CA16107.

## 2. Phytosanitary-Regulated *Xanthomonas* in the European Union

The European Commission, following opinions published by the European Food and Safety Authority (EFSA) Panel on Plant Health, changed the status of several *Xanthomonas* from quarantine species present but not widespread pest in the EU to Regulated Non-Quarantine Pests (RNQP) (EU Regulation 2016/2031 and Commission Implementing Decision 2019/2072). Practically speaking for quarantine pests, no presence is accepted in the material being moved (the plant material must be free from the pest). An RNQP is a pest with a well-established identity, present in a country, and mainly transmitted via plants for planting. Additionally, its presence on plants for planting affects the intended use of those plants with an unacceptable economic impact. For these organisms, feasible and effective measures are available to prevent their presence on the plants for planting concerned. Unlike for quarantine pests, tolerances of presence may be accepted for RNQPs. However, this is not the case for xanthomonads.

Laboratory testing of plants and plant products during official controls, i.e., following mandatory phytosanitary regulations and procedures authorized by National Plant Protection Organizations (NPPOs), is based on official protocols for most of the regulated *Xanthomonas*. Official diagnostic protocols in plant health are standards describing procedures and methods for the diagnosis (i.e., detection and identification) of pests, which have been compiled by expert scientific committees of internationally recognized organisations in plant health, to address the need for harmonization of the way plants, plant products, or other regulated articles are examined worldwide for the possible presence of these pests [39,40,41]. According to the Regulation (EU) 2017/625 they could be developed or recommended by the EU Reference Laboratories, evaluated via inter or intra-laboratory validation studies, and wherever possible, characterized by the relevant core performance criteria, i.e., analytical sensitivity (limit of detection), analytical specificity (inclusivity and exclusivity), repeatability, reproducibility, and accuracy. These protocols can be used by laboratories authorized by the NPPOs in such a manner that the results of the pest diagnosis may be considered when deciding on a phytosanitary measure. Furthermore, they enhance the mutual recognition of diagnostic results by NPPOs facilitating trade and aid the development of expertise and technical cooperation [39,40,41]. These protocols can be sourced from the open-access websites of organisations such as EPPO and Food and Agricultural Organization of the United Nations (FAO) via IPPC.

**Table 1 microorganisms-09-00862-t001:** Diseases, their regulation and recommendations for management.

Disease	Pathogen	EPPOList	EPPO Diagnostic Protocol	Commission Implementing Regulation (EU) 2019/2072	EFSA Documents
Citrus bacterial canker	*X*. *citri* pv. *aurantifolii*; *X*. *citri* pv. *citri*	A1	[42]	A1 Quarantine pest (Annex II A)	[43,44]
Onion bacterial blight	*X*. *euvesicatoria* pv. *allii*	A1	[45]		
Bacterial leaf blight of rice	*X*. *oryzae* pv. *oryzae*	A1	[46]	A1 Quarantine pest (Annex II A)	[47]
Bacterial leaf streak of rice	*X*. *oryzae* pv. *oryzicola*	A1	[46]	A1 Quarantine pest (Annex II A)	[47]
Bacterial blight of hazelnut	*X*. *arboricola* pv. *corylina*	A2	[48]	RNQP (Annex IV)	
Bacterial spot of stone fruits	*X*. *arboricola* pv. *pruni*	A2	[49]	RNQP (Annex IV)	[50]
Bacterial blight of walnut	*X*. *arboricola* pv. *juglandis*	NA	NA	RNQP (Annex IV)	
Bacterial leaf spot of poinsettia	*X*. *axonopodis* pv. *poinsettiicola*	A2	NA		
Bacterial spot of tomato and sweet pepper	*X*. *euvesicatoria* pv. *euvesicatoria*	A2	[51] *	RNQP (Annex IV)	[52]
	*X*. *euvesicatoria* pv. *perforans*	A2	[51] *	RNQP (Annex IV)	
	*X*. *hortorum* pv. *gardneri*	A2	[51] *	RNQP (Annex IV)	
	*X*. *vesicatoria*	A2	[51] *	RNQP (Annex IV)	
Bacterial angular leaf spot of strawberry	*X*. *fragariae*	A2	[53] *	RNQP (Annex IV)	
Bacterial blight of anthurium and other aroids	*X*. *phaseoli* pv. *dieffenbachiae*	A2	[54] *		
Common bacterial blight of bean	*X*. *phaseoli* pv. *phaseoli*	A2	NA	RNQP (Annex IV)	[55]
Bacterial leaf streak and black chaff of small-grain cereals	*X*. *translucens* pv. *translucens*	A2	NA		
Leafspot and dieback of ornamental fig	*X. campestris* pv. *fici*		NA	RNQP (Annex IV)	

NA: not applicable; * protocol under revision or revision planned. International Seed Testing Association (ISTA), International Seed Federation (ISF), United States Department of Agriculture (USDA), and National Seed Health System (NSHS). Other sources of reviewed detection methods include scientific publications of EFSA (‘Scientific Opinions’, ‘Pest Survey Cards’, etc) and other peer-reviewed scientific journals.

As for other pest diagnostic protocols, *Xanthomonas* spp. protocols include a description of symptoms, sampling procedures of plants and plant products, methods for detecting the pest in a commodity, methods for extracting, isolating, identifying the pest from plant tissues, as well as sources to confirm its pathogenicity on host plants (Appendix A). They usually include more than one method, often presented sequentially in flow diagrams, to consider the capabilities of laboratories and the circumstances of use (e.g., symptomatic or asymptomatic plant tissues, type of plant tissue). The methods included are selected based on scientific literature data regarding their sensitivity, specificity, and reproducibility, as well as expert judgement. Currently, official protocols for the regulated *Xanthomonas* species were published between 2005 and 2017. Therefore, they may include some techniques that are less and less used (Appendix A). Official diagnostic protocols provide at least the minimum requirements for reliable diagnosis and are subject to review and amendment considering new developments in the field. They can be adjusted by individual laboratories if the adjustments are adequately validated [39,40,41]. In general, it is recommended that diagnosis is based on at least two tests, based on a different biological principle or by molecular tests that target different parts of the genome.

## 3. Molecular Methods in Diagnosis and Detection of Regulated Xanthomonads

Bacterial diseases are notoriously difficult to control and require an intensive integrated management approach to mitigate serious economic losses. Control includes cultural practices, bactericide, or plant defence inducers and where applicable, plant resistance and biocontrol strategies [56]. The effectiveness of integrated disease management strategies strongly depends on the availability of rapid, sensitive, and specific diagnostic methods. Diagnostic methods are also of pivotal importance in preventing phytopathogenic bacteria introduction into nurseries and cultivation areas, thus contributing to disease control by exclusion [57,58].

The term ‘diagnosis’ generally refers to the identification of the nature and cause of a disease problem and therefore is related to the presence of a plant with symptoms [59]. Conversely, ‘detection’ is used when establishing the presence of a target pathogen within a sample [60], e.g., a plant or part of it, even in the absence of symptoms, as well as in vectors or environmental samples such as soil or water. Therefore, detection may imply a low bacterial titre requiring—together with analytical specificity—high analytical sensitivity [57].

Methods based on the analysis of nucleic acids have brought with them a change to the diagnosis of plant diseases improving the ability to quickly detect, identify and characterize disease agents. They need to meet certain criteria, the most important being the analytical specificity that not only comprises the exclusive detection of the suspected pathogenic bacteria (inclusivity) but also includes avoiding the misidentification of non-pathogenic bacteria or closely related but non-target ones (exclusivity). Besides, the method must be sensitive enough to detect bacteria when they are present at a low level, for example in asymptomatic samples, or even from plant material containing inhibitors that make DNA difficult to be amplified. Finally, molecular methods used routinely must be carefully designed and simple enough to be repeatable and reproducible, being affordable for routine diagnostic labs.

DNA markers allow the detection of xanthomonads in a specific and sensitive manner. In the past, taxon-specific markers were identified either based on previous knowledge of gene function (e.g., quinate metabolism, *hrp*-related, pilus assembly, ribosomal, and housekeeping genes) or using fingerprinting methods such as random amplified polymorphic DNA (RAPD). However, the accumulation of genomic information in recent years has facilitated the identification of new DNA markers using in-silico comparative genomics approaches. Several possible pipelines for the identification of taxon-specific markers were suggested [61,62,63,64]. In general, these pipelines rely on two main steps. The first step is the identification of sequences that are highly conserved in the taxon of interest, either using BLAST or other online resources that contain information on taxon-specific genes [62]. In the second step, the resulting sequences are aligned against public databases to identify taxon-specific sequences. Furthermore, sequences can be filtered based on additional criteria, for example, to avoid plasmids and regions that are close to transposable elements [63]. The molecular markers are transversal across the various detection methods, being specific probes designed according to the selected method. In some cases, the same DNA marker is used for the development of various detection methods. More information on Genomics-Informed Approaches for Molecular Diagnostics can be found in Section 5. Below, we discuss several detection methods, including hybridization-based, PCR-based and LAMP and other isothermal methods.

### 3.1. Hybridization-Based Methods

Hybridization methods, which include microarrays, membrane-hybridization (or macroarrays), and fluorescence in situ hybridization (FISH), were originally developed to detect and quantify transcripts, to measure gene expression, or to track single bacterial cells by fluorescence microscopy in complex matrices [65,66]. The advent of microarrays in the early 2000s, and the possibility to screen dozens to thousands of genomic markers simultaneously, including both taxa-specific markers and markers targeting genes coding for functional traits, raised the optimism for making microarrays capable of screening a broad range of pathogens simultaneously [67]. The technological resources required for microarray printing, hybridization and signal detection, and the complex and time-consuming interpretation of the data, hindered further developments towards the use of microarray hybridization technology for pathogens detection. In the last decade, dot-blot hybridization using nylon membranes has been extensively used to detect and identify diverse animal pathogens [68,69] and plant pathogens including different *Xanthomonas* species [32,62,70,71]. In some of these studies, several DNA markers have been combined in a multiplexing technique to detect and discriminate closely related *Xanthomonas* species and even pathovars. Using an array of 21 *Xanthomonas*-specific DNA markers, it was possible to observe species-specific hybridization patterns and distinguish *X. fragariae*, *X. phaseoli* pv. *phaseoli* and *X. citri* pv. *fuscans* [70]. The same approach was described to detect *X. euvesicatoria* in tomato and pepper plants [62]. More recently, a protocol of dot-blot hybridization using nine DNA markers has been proposed to identify *X. arboricola* pv. *juglandis* [32]. The strain-dependent hybridization patterns that were observed led the authors to suggest that these patterns could be informative for distinguishing *X. arboricola* pv. *juglandis* strains and used in combination with Multilocus Sequence Analysis (MLSA) to characterize the diversity of this pathovar in a broad study [32].

A major concern of hybridization-based methods to detect DNA markers has been the reproducibility and the interpretation of weak dot-blot signals. Nevertheless, the selection of highly specific DNA markers with lengths of a few hundred base pairs (bp) and hybridization carried out under high-stringency conditions may increase the consistency of dot-blot hybridization signals obtained [72]. In addition, an automatic analysis can be performed to overcome the operator subjectivity.

### 3.2. PCR-Based Methods

During the past 25 years, PCR-based methods have become the gold standards for diagnosis of plant bacterial diseases—those caused by xanthomonads being no exception. PCR techniques have a substantial advantage over other methods as they do not require pathogen isolation and cultivation and are considered more sensitive and less time-consuming than culture-based methods [73].

Most of the techniques for regulated xanthomonads were initially developed for conventional endpoint PCR and, afterwards, were adapted, or new protocols designed, for quantitative real-time PCR (qPCR). In qPCR, amplification and detection of amplified products are coupled in a single reaction vessel, eliminating the need for post-amplification processing. In addition, analysis of the amplification reaction kinetic allows for quantification of the starting templates in a sample, and therefore pathogens cannot only be detected but can also be quantified. These two features, alongside the higher sensitivity compared to endpoint PCR, have made qPCR one of the most important tools lately implemented in plant protection service laboratories.

Recently, digital PCR (dPCR) was developed based on the division of the sample into numerous partitions, which are amplified individually. This tool presents some advantages over qPCR for diagnostics, and it is associated with higher sensitivity, a critical point for the detection of regulated organisms [74,75]. A summary of PCR-based protocols for phytosanitary regulated *Xanthomonas* spp. is compiled in Table 2 and summarized below.

For *X. euvesicatoria* pv. *allii*, the causal agent of leaf blight of onion, a multiplex nested PCR was developed based on pilus assembly genes, *pilW* and *pilX*, and the avirulence gene *avrRxv*, originally identified as specific for this bacterium by RAPD and amplified fragment length polymorphism (AFLP) [76]. In addition, protocols for qPCR were developed based on the same genes [77].

A similar development occurred for citrus bacterial canker, caused by *X. citri* pv. *citri* and *X. citri* pv. *aurantifolii* types B and C. Extensive work has been undertaken on citrus canker diagnosis to compare and set up the most appropriate protocols to detect and identify strains of the three citrus canker types [78,79,80]. Protocols were designed for targeting genes of ribosomal sequences or those identified to be factors involved in virulence such as *rpf*, *hrpW,* or *pthA*. Besides, the precise identification of new targets was recently performed based on comparative genomic analysis of the different types of xanthomonads that cause disease in citrus [80]. Several protocols were initially developed on conventional PCR [81,82,83,84,85,86] or recently into droplet PCR [75].

Detection of *X. arboricola* pathovars is achieved by different protocols, both based on PCR of specific target genes identified after genotyping analysis and comparative genomic analysis. Specific virulence factors such as *xopE3* and *hrpD2* (*hrcR*) have been used in addition to an ABC transporter gene to detect and identify *X. arboricola* pv. *pruni*. Primers based on these genes have been used in conventional PCR or qPCR, following simplex or multiplex approaches, allowing to distinguish between pathogenic and non-pathogenic *Xanthomonas* strains isolated from *Prunus* spp. [31,87,88,89,90]. For *X. arboricola* pv. *juglandis*, a thorough in silico analysis was performed to select specific sequences from the bacteria that were characterized and used to develop PCR and qPCR protocols [32,91]. Finally, on *X. arboricola*, a duplex PCR assay, using primers designed and based on *ftsX* and *qumA* genes, were used to identify *X. arboricola* pv. *corylina* [90,92].

PCR-based techniques proposed for bacterial blight and the bacterial leaf streak of rice, caused by *X. oryzae* pv. *oryzae,* and *X. oryzae* pv. *oryzicola*, respectively, are also numerous, providing tools to accurately detect both pathogens based on *hrp* or ribosomal sequence genes, a *rhs* family gene and a gene encoding a putative glycosyltransferase for pv. *oryzae*, and a membrane fusion protein gene and *avrRxo1* gene for pv. *oryzicola*. These protocols were designed to detect those bacteria individually or together with other rice pathogens in a multiplex approach [61,93,94,95,96,97,98,99].

Diagnosis of bacterial spot of tomato and pepper has been difficult because the causative origin is due to four different bacteria: *X. euvesicatoria* pvs. *euvesicatoria* and *perforans*, *X. hortorum* pv. *gardneri*, and *X. vesicatoria*. Several PCR approaches have been developed to detect and identify the specific strains causing the disease, using primers for *rhs* family, housekeeping genes and recently, a coding sequence of a zinc-dependent oxidoreductase [100,101,102,103,104,105].

**Table 2 microorganisms-09-00862-t002:** Main protocols described for regulated xanthomonads based on amplification of specific target DNAs.

Disease	Bacteria	Conventional PCR	qPCR ^1^	IA ^2^ Methods
Onion bacterial blight	*X. euvesicatoria* pv. *alli*	[76]	[77]	NA ^3^
Citrus bacterial canker	*X. citri* pv. *citri* and *aurantifolii* pathotypes B and C	[81,82,83,84,85,86]	[80,106,107]	[108]
Bacterial spot of stone fruits, walnut blight, hazelnut blight	*X. arboricola* pvs. *pruni*, *corylina* and *juglandis*	[32,87,90,92]	[31,88,91]	[109,110]
Bacterial leaf blight and bacterial leaf streak of rice	*X. oryzae* pvs. oryzae and *oryzicola*	[61,93,94,95,97,98,99]	[95,96]	[111]
Bacterial spot of tomato and sweet pepper	*X. euvesicatoria* pvs. *euvesicatoria* and *perforans*, *X. hortorum* pv. *gardneri*, *X. vesicatoria*	[100,101,102,103,105,112]	[104,105]	[113,114,115]
Bacterial angular leaf spot of strawberry	*X. fragariae*	[116,117,118,119]	[120,121,122]	[63,123,124,125]
Bacterial blight of anthurium and other aroids	*X. phaseoli* pv. *dieffenbachiae*	[126,127,128]	[90]	[129]
Bacterial leaf spot of poinsettia	*X. axonopodis* pv. *poinsetticola*	[130]	NA	NA
Bacterial leaf streak	*X. translucens* pv. *translucens*	[131]	NA	[132]
Common blight of bean	*X. phaseoli* pv. *phaseoli, X. citri* pv. *fuscans*	[133]	NA	[134]

^1^ qPCR: quantitative real-time PCR; ^2^ IA: isothermal amplification: ^3^ NA: not applicable.

Diagnosis of bacterial angular leaf spot of strawberries is especially difficult by bacterial isolation since *X. fragariae* are not easily cultured and a high number of bacteria are not usually expected in strawberry, particularly in asymptomatic plants. A few PCR and qPCR protocols were designed, some of them including an additional step of immunocapture or bioenrichment to increase bacterial concentration in the sample before amplification [116,117,118,119,120,121,122].

For anthurium bacterial blight, caused by *X. phaseoli* pv. *dieffenbachiae*, protocols for conventional PCR or qPCR were developed based on primers designed from genotyping analysis and sequence characterized amplified regions (SCAR), sometimes using a preceding immunocapture step [90,126,127,128]. Meanwhile, for the diagnosis of bacterial leaf spot poinsettia, a PCR protocol has been described targeting specific sequences identified after a comparative genomic analysis among *X. axonopodis* pv. *poinsettiicola*, *X. hyacinthi* and *X. campestris* pv. *zantedeschiae* [130].

Finally, for bacterial leaf streak, caused by pathovars of *X. translucens*, a PCR was designed to detect and discriminate cereal-pathogenic xanthomonads in seeds based on ribosomal internal transcribed spacer (ITS) variability [131], and for bacterial blight of bean, caused by *X. phaseoli* pv. *phaseoli* and *X. citri* pv. *fuscans*, PCR primers were designed based on a SCAR region from a specific plasmid sequence [133]. In the Commission Implementing Regulation (EU) 2019/2072, *X. campestris* pv. *fici* is also mentioned as a regulated pathogen for fig trees; however, no specific PCR-based test is available. Leite et al. [135] used primers targeting *hrp* genes to detect this pathogen, but they are not specific for this pathovar.

Different procedures for extracting nucleic acids from different types of plant specimens, such as leaves and seeds, have been described for xanthomonads detection, providing DNA samples of suitable quality and quantity. Nevertheless, these samples may contain substances that inhibit PCR processes and negative results would not necessarily indicate the absence of the pathogen but an unsuccessful amplification. Inhibitor’s presence can be identified by monitoring the amplification of a second target nucleic acid, which serves as an internal control (IC). Two different types of ICs have been developed for xanthomonads detection. A ribosomal sequence from the plant present in all specimens was co-amplified together with the target sequence in the protocols developed for *X. euvesicatoria* pv. *alli* and *X. citri* pv. *citri* [76,77]. Besides, another IC control described for *X. citri* pv. *citri*, included a synthetic IC consisting of a plasmid with primer binding regions identical to those of the target sequence and a randomized internal sequence different to it in length [107]. Both models ensure amplification of at least one product in each reaction and if this does not occur, the test result cannot be interpreted. Moreover, if amplification is delayed (higher C_q_ values), this would imply inhibition due to DNA binding [136].

False-positive and false-negative results, as occurred for other pathogens, are avoided in xanthomonads by proper primer design, the use of appropriate DNA extraction protocols and when available, using ICs. However, another aspect must be considered: it is impossible to determine by PCR whether the xanthomonad detected is viable and infective because no information about bacterial cell integrity can be deduced using standard protocols. Detection of viable cells may be an important feature, particularly in the case of regulated organisms, given its implication in epidemiological and risk assessment aspects and this will be discussed in Section 3.4.

### 3.3. LAMP and Other Isothermal Methods

Isothermal amplification (IA) methods operate at a uniform temperature without the need for variation during the process, therefore eliminating the use of a thermocycling machine [137]. Several IA methods have been developed in the last two decades encompassing Nucleic Acid Sequence Based Amplification (NASBA), Sequence Mediated Amplification of RNA Technology (SMART), Strand Displacement Amplification (SDA), Recombinase Polymerase Amplification (RPA), Loop-mediated Isothermal Amplification (LAMP) and Multiple Cross Displacement Amplification (MCDA) (for reviews see [137]). Although IA methods differ in features such as the number of primers and enzymes, the temperature of amplification, and template types used, they all share some common features. For example, since the strands of DNA are not heat-denatured, all isothermal methods rely on a polymerase with strand-displacement activity to enable primer binding and initiation of the amplification reaction.

Since IA methods provide detection of a nucleic acid target sequence in a streamlined and exponential manner, they are often used for diagnostics. As nucleic acid targets are amplified in such a high amount in a short time compared to PCR, they can be detected by measuring turbidity or by naked eye inspection for colour change. However, to avoid any result misinterpretation, real-time readings of nucleic acid target amplification by these methods are also possible and several portable instruments are commercially available. This capability thus eliminates the cumbersome need for gel electrophoresis rendering it even easier to use. IA methods are also more tolerant to inhibitors present in crude samples than conventional and qPCR, thus simplifying sample preparation. Additionally, since IA methods are using multiple primers, analytical specificity increases compared to PCR. All the aforementioned reasons render IA methods very promising for on-site diagnostics such as in-field or entry points (e.g., airports or piers) detection.

Diagnostics of regulated xanthomonads using IA methods mainly rely so far on LAMP, one study on NASBA and another one on RPA (Table 2). So far, IA methods for xanthomonads diagnostics were all developed on DNA and none on RNA. The identification of specific nucleic acid target sequences is mainly informed by comparative genomics (see Section 5) and few studies are either based on sequence alignments (e.g., *recG*) or targets identified previously with other techniques (e.g., RAPD or SCAR markers). The first IA methods developed for *Xanthomonas* species was with the detection of the seed-borne pathogen *X. hortorum* pv. *carotae* in carrot seed using LAMP [138]. However, the responsible agent of bacterial spot of stone fruits, *X*. *arboricola* pv. *pruni*, was the first regulated xanthomonad for which an IA method based on LAMP was developed [109].

The rapidity of the amplification, the tolerance to inhibitors, the limited equipment and resource requirements, the on-site capacities as well as the possibility to derive these methods into viability assays (see Section 3.4) renders these methods attractive for the diagnostics of xanthomonad. For these reasons, IA methods are progressively being introduced into official diagnostics schemes for regulated xanthomonads such as the ones from EPPO and IPPC.

### 3.4. Viability PCR

Current methods (serology, conventional PCR or qPCR, LAMP and other isothermal methods) do not allow assessing the viability of target organisms. The culture-based methods have limitations mainly due to the competition with saprophytic flora, the lack of specificity and the slow growth rate of some xanthomonads (i.e., up to seven days for *X. fragariae*). Determining whether a pathogen is viable is of critical importance in assessing the potential biological risk, especially in the case of regulated pathogens.

Membrane integrity is an accepted biomarker for discriminating viable cells because cells with compromised membranes are already dead or nearly so [139]. Combining PCR with DNA intercalating dye (viability PCR or vPCR) is an efficient method to determine the viability of cells based on membrane integrity [140]. The dye penetrates only dead cells with compromised membranes and intercalates covalently into the DNA after photoactivation, subsequently interfering with DNA amplification.

The first vPCR assay was developed by Novga et al. [141] using ethidium monoazide (EMA) and later Nocker et al. [142] proved the efficiency of the alternative molecule propidium monoazide (PMA). The use of PMA is generally favoured today because although EMA is slightly more effective in suppressing PCR in dead cells, it is capable of penetrating living cells and generating false-negative results [143]. The first xanthomonads vPCR was developed for the seed-borne pathogen *X. hortorum* pv. *carotae* in carrot seed [138]. PMA treatment was combined with TaqMan qPCR assay or LAMP to successfully detect viable cells of *X. hortorum* pv. *carotae* cells in commercial carrot seed lots before and after hot-water treatment. A live/dead cells distinction protocol using PMA coupled with qPCR or LAMP was also developed for analysing seed washes for the detection of a specific subset of *X. vasicola* pv. *vasculorum* strains (*X. campestris* pv. *zeae*-like strains). Both PEMAX-PCR (a mix of PMA and EMA) and PMA-qPCR have been recently developed for the detection of viable cells of *X. fragariae* in strawberry [124,125].

These viability protocols allow for the accurate detection and quantification of viable cell populations in plants or seed. These assays can assist in the reliable detection of infected planting material and could help timely decisions to be made at the border on imported materials. It is important to develop viability protocols for regulated xanthomonads pathogens to be included in diagnostic standards to satisfy quarantine regulations. Furthermore, these viability molecular tools can also support ecological (i.e., the importance of Viable But Not Cultivable (VBNC) state) and in planta fitness studies.

## 4. Taxonomy and Diversity of Xanthomonads

Having a sound knowledge of the diversity of groups of strains is a prerequisite for accurate testing of plant material for trade, the study of plant disease epidemiology and the implementation of control methods. The identification of strains at the genus level relies on DNA sequencing of the 16S rRNA gene, which has, however, a low predictive value to differentiate groups of strains at the infraspecific level within various genera [144] including the genus *Xanthomonas*. A range of phenotyping methods can be used to characterize the diversity of xanthomonads. Pathogenicity profiling is one means to capture the diversity of *Xanthomonas* spp. and is an essential contribution to the description of novel pathovars [17,145]. Phytopathogenic specialization is, however, not correlated to the phylogeny, as several pathovars are polyphyletic [146]. Moreover, within a pathovar, groups of strains may interact with variants of host species. This is the case for pathovars *campestris*, *glycines*, *malvacearum*, *vesicatoria*, and *oryzae*, which harbour races interacting with host varieties carrying specific resistance genes [6]. Pathogenicity tests are costly and time-consuming, and their extent is limited to a few plant species that can be tested simultaneously [146]. Profiling of carbon source utilization, whole-cell proteins, fatty acid methyl esters and serological properties are also used to describe the diversity at a specific and infraspecific level and contribute to species description in a polyphasic approach [147,148].

DNA profiling exploits the natural genetic variation present in DNA to provide molecular genetic markers capable of identifying, differentiating, and characterizing organisms. Numerous fingerprinting methods based on restriction fragment length polymorphism (RFLP), AFLP, arbitrarily-primed PCRs, including rep-PCR [149] and RAPD were used before the advent of genomics to study the diversity of various *Xanthomonas* groups of interest [150,151,152,153,154]. The main pitfalls of these fingerprinting methods are their limited repeatability and reproducibility among laboratories, in contrast to the most recent molecular methods based on DNA sequencing. Whilst bacterial identification moved to single-locus sequencing, MLSA has become the most popular molecular method to establish the phylogenetic relationships between bacterial species, including *Xanthomonas* spp. [155,156,157]. This method derives from the multilocus sequence typing (MLST) method proposed by Maiden et al. [158], which is widely used for epidemiological purposes. However, for those groups of bacteria that are mostly clonal and do not present sufficient diversity to be exploited through the analysis of housekeeping gene sequencing, other methods based on rapidly evolving markers similar to microsatellites, such as VNTR, can be useful. Other molecular markers, such as CRISPRs that are, however, not widely distributed with the genus *Xanthomonas*, could be useful for specific taxa. Genome-based methods are now the gold standard to accurately study *Xanthomonas* diversity and refine the taxonomy of groups, the contours of which were not precisely defined, as was the case for the species *X. cynarae, X. gardneri,* and *X. hortorum* [13].

### 4.1. Markers for Taxonomy and Phylogeny

#### 4.1.1. Single-Locus Diagnostics

Bacterial identification, which was traditionally addressed by morphological, physiological, serological and/or biochemical examinations was revolutionized with the advent of DNA sequencing. Highly conserved loci, such as the ribosomal RNA genes, were among the first to be targeted. Portions of the 16S rRNA gene proved to be valuable in assigning the genus to a bacterial isolate. In 1997, Swings and co-workers performed the first comprehensive analyses of 16S rRNA gene sequences from *Xanthomonas* and a related genus, *Stenotrophomonas*, which allowed distinguishing these two genera and identified 20 species of *Xanthomonas* [159,160]. Sequences from the two genera differed by 50 nucleotide positions on average, whereas species of *Xanthomonas* differed by only 14 nucleotides on average. Due to the very restricted variability in 16S rRNA gene sequences within the genus *Xanthomonas*, DNA signatures with useful diagnostic value for differentiating the *Xanthomonas* species could not be identified. Indeed, the restricted diagnostic value of the 16S rRNA gene sequences led to several spurious misidentifications of xanthomonads from various specimens, such as respiratory secretions of cystic fibrosis patients [161], skin microbiota of dermatitis patients [162], the holobiome of host-seeking ticks and mosquitos [163,164], or Chinese permafrost soils [165].

Later, intergenic or protein-coding sequences were targeted for diagnostics because of their larger degree of sequence variability, among them the 16S-23S intergenic spacer and the *lrp*, *gyrB* and *rpoB* genes [166,167,168,169]. The *gyrB* gene proved to have high diagnostic potential and helped to identify the causal agents of new diseases [170,171,172,173,174,175,176,177] and to clarify the phylogenetic structure of the genus *Xanthomonas* [178].

#### 4.1.2. Multilocus Sequence Analysis/Multilocus Sequence Typing

To improve the diagnostic value of purely DNA sequence-based approaches and to minimize the confounding effects of interspecies recombination, several partial gene sequences were combined, leading to MLSA and MLST [179,180,181,182]. In parallel, specific databases were developed allowing comparison between own isolates and the wealth of characterised strains of worldwide origin [183,184,185,186].

For xanthomonads, three MLSA schemes have evolved in a short time and have been used by different ‘schools’ of epidemiologists. The first version came from a French team that had included *atpD*, *dnaK*, *efp,* and *glnA* in their scheme [187], which later was supplemented by the *gyrB* gene [188]. The second scheme was developed in New Zealand and included *dnaK*, *fyuA*, *gyrB,* and *rpoD* [157]. The third scheme was established in the United States of America, targeted *fusA*, *gapA*, *gltA*, *gyrB*, *lacF*, and *lepA* [189]. Even if all three schemes have one gene in common, *gyrB*, they, unfortunately, targeted different regions of the gene and data from one scheme cannot be compared with data from another scheme. Concomitant with the last scheme, the PAMDB database was developed (http://pamdb.org), which serves as a repository for all three schemes.

MLSA is highly congruent with other typing methods, such as AFLP or MultiLocus Variable number of tandem repeats Analysis (MLVA, as explained below), but with much lower resolution [24,190,191,192,193]. On the other hand, MLSA proved to be more efficient than fingerprint methods, e.g., rep-PCR, for large populations and in comparisons at the global scale [194,195,196]. MLSA allowed comparing strains from different species and contributed thus to a better taxonomic framework for the genus *Xanthomonas* [157]. Because of its robustness and user-friendliness, MLSA has often been used to identify new pathogens, e.g., pathogens of avocado, eucalypt, lavender, nectarine, peony, radicchio, or roses [197,198,199,200,201,202,203].

#### 4.1.3. MultiLocus Mass Typing Using MALDI-TOF Mass Spectrometry

If DNA-based procedures have revolutionized bacterial identification, matrix-assisted laser desorption ionization (MALDI) time-of-flight (TOF) mass spectrometry (MS) has increasingly replaced these procedures for the quick and reliable identification of bacteria especially in clinical microbiology laboratories [204]. MALDI-TOF MS indeed offers the possibility of accurately resolving bacterial identity to the genus, species, and subspecies levels in some taxa [205,206]. This method is a type of mass spectral fingerprinting in which sequence variation of housekeeping proteins such as ribosomal proteins was shown in several bacterial species to produce peaks shift on the MALDI mass spectra that can then be used as biomarkers for identification and turning thus the analysis into a multilocus mass typing (MLMT) of markers spread over the genome [205,206,207].

The most streamlined approach consists of direct MALDI-TOF MS analysis, in which a single colony from bacteria grown on agar in a Petri dish is deposited directly onto a sample target before the addition of a matrix to lyse and release the intracellular proteins [208]. A fingerprint is then obtained after ionization and separation according to the mass-to-charge ratio (*m*/*z*) of these proteins. Such MALDI-TOF MS profiles can then be used to identify unknown bacterial samples down to the subspecies level, providing they can be matched against comprehensive databases of empirical spectra or in silico predicted protein molecular masses [205,206,207].

MALDI-TOF MS identification has proven to be successful with several *Xanthomonas* species, of which several are regulated such as *X. fragariae,* where it could distinguish this regulated pathogen from *X. arboricola* pv. *fragariae* strains, or such as *X*. *oryzae* where discrimination of the two rice pathovars was demonstrated [209,210,211,212]. However, the lack of reference profiles for these phytopathogenic bacteria in commercially available databases often hinders their identification, a gap filled by coordinated efforts to target these organisms [213] or by in silico predicted protein molecular masses-based databases derived from the ever-increasing wealth of genome sequence data [205,206].

### 4.2. Markers for Population/Epidemiology Studies

#### 4.2.1. MultiLocus Variable Number of Tandem Repeats Analysis

Tandem repeats (TR) are small pieces of satellite DNA presented as ≥2 perfect or imperfect (i.e., showing degeneracy) copies arranged in a head-to-tail manner and present in coding and non-coding regions of genomes [214]. TRs are commonly designated as microsatellites, minisatellites, or satellites according to their size, although there is no general agreement on boundaries defining these three TR classes [214]. Two main non-exclusive mechanisms, slipped strand mispairing (SSM) and recombination, are involved in the size variation of TR arrays [215,216]. The extensive polymorphism at microsatellite arrays primarily is the consequence of SSM during DNA replication [217]. A single repeat unit is preferentially added or deleted during SSM and polymorphism can be best modelled using generalized stepwise mutation or two-phase models [218,219]. Polymorphism can also arise from recombination of whatever the repeat unit size is, but this mechanism prevails for arrays composed of large repeat units [216]. Bacteria exploit the instability of TRs in coding regions to reversibly regulate the expression of specific genes, a feature that contributes to their adaptation against changing environments [216].

The first TR-based MLVA (for multilocus variable number of tandem repeats analysis) scheme used for subtyping a bacterial species was developed for *Haemophilus influenzae* [220]. Since then, MLVA proved useful for tracing many foodborne and human or animal bacterial pathogens [221,222]. The wide availability of complete or near-complete bacterial genomic sequences and numerous dedicated bioinformatic tools has markedly facilitated the detection of TRs and the development of new MLVA schemes [223,224,225,226,227]. The first MLVA scheme developed for a plant-pathogenic bacterium targeted *Xylella fastidiosa* [228] and the first one targeting a xanthomonad was developed a few years later for *X. citri* pv. *citri* [229]. Many MLVA assays have been described for xanthomonads since these pioneering studies (Table 3; Appendix A).

Minisatellites (herein defined as TRs ranging from 10 to 250 bp in size) and microsatellites (herein defined as TRs <10 bp in size) have been used for subtyping bacteria at an infrasubspecific level. First largely implemented for *Mycobacterium tuberculosis* [230,231], minisatellite typing was found useful for the global epidemiology of *X. citri* pv. *citri*, as this technique combines technical easiness and a good phylogenetic signal congruently matching single nucleotide polymorphism (SNP) data derived from whole-genome sequencing and accurately identifying the three pathogenic variants (i.e., pathotypes) delineated within this pathovar [33,232,233].

**Table 3 microorganisms-09-00862-t003:** Tools for molecular typing of regulated *Xanthomonas*.

Pathogen	Fingerprints	VNTR/MLVA	CRISPR	MLSA/MLST
**A1 list**				
*X. citri* pv. *aurantifolii*	NA ^1^	NA	NA	[234]
*X. citri* pv. *citri*	[190,235]	[229,232,236]	[237]	[190,234]
*X. euvesicatoria* pv. *allii*	[238,239,240,241]	[242]	NA	[234]
*X. oryzae* pv. *oryzae*	[243]	[244]	NA	[245]
*X. oryzae* pv. *oryzicola*	NA	[244,246]	NA	[245,247]
**A2 list**				
*X. arboricola* pv. *corylina*	NA	[24,248]	NA	[24]
*X. arboricola* pv. *juglandis*	[154,249,250]	[24]	NA	[24]
*X. arboricola* pv. *pruni*	[189]	[24,250]	NA	[24,187]
*X. axonopodis* pv. *poinsettiicola*	NA	NA	NA	[251]
*X. euvesicatoria* pv. *euvesicatoria*	[191,252]	[242]	NA	[191,234,252,253]
*X. euvesicatoria* pv. *perforans*	[191,252]	[242]	NA	[191,252]
*X. fragariae*	[254,255]	[192]	[192]	[192]
*X. hortorum* pv. *gardneri*	[191,252]	NA	NA	[191,252]
*X. phaseoli* pv. *dieffenbachiae*	[256,257]	NA	NA	[234]
*X. phaseoli* pv. *phaseoliX. citri* pv*. fuscans*	[258]	NA	NA	[234]
*X. translucens* pv. *translucens*	[259,260]	NA	NA	[261,262,263]
*X. vesicatoria*	[191,254]	NA	NA	[191,252,253]

^1^ NA: not available.

Because of its high discriminatory power, microsatellite typing has been widely used for epidemiological analyses of genetically related strains at small to medium spatiotemporal scales. In contrast, because microsatellites are characterized by high levels of size homoplasy, these markers are not suitable for precisely assessing deep genetic relatedness among populations, and in such cases, phylogenetic analyses of SNP data from non-recombinant genomic regions should be preferred. Some genotyping schemes combining both marker classes have sometimes been proposed and aimed at providing high discriminatory power and an improved phylogenetic signal [192,249,264,265]. Recent developments include direct microsatellite typing from diseased plant material using a method that does not require cultivating the target bacterium nor purifying genomic DNA [266].

Microsatellite typing allowed gaining knowledge on the population biology of several xanthomonads. More specifically, it was shown useful for discriminating between genetically-related populations differing in pathogenicity [24], deciphering the geographical structure of *Xanthomonas* populations at scales ranging from a single field to a production basin and placing hypotheses on source populations [26,192,193,246,264,267,268,269,270], testing approximate Bayesian computation-based invasion scenarios [236], emphasizing the importance of plant propagative material as the source of disease emergence [236,265,271,272], estimating the range of spatial dependency of outbreaks and assessing the biological significance of coinfections in single lesions [266].

#### 4.2.2. Clustered Regularly Interspaced Short Palindromic Repeat Genotyping

Clustered regularly interspaced short palindromic repeats (CRISPR) represent another class of molecular markers that have been exploited for molecular typing of several bacterial species, such as mycobacteria, legionellae, and salmonellae [273,274]. The corresponding genotyping method, called spoligotyping (for spacer oligonucleotide typing), is based on the detection of unique spacers in the CRISPR locus. CRISPR analyses are of special interest because they can easily provide a chronological perspective on the ancestry and genealogy of bacterial isolates once the evolution of this locus in a bacterial species has been inferred [237]. Remarkably, the presence of CRISPR systems appears to be conserved at the infraspecific level, i.e., within pathovars of a given *Xanthomonas* species. Important xanthomonads with CRISPR loci include *X. albilineans, X. campestris* pv. *raphani*, *X. cassavae*, *X. citri* pv. *citri*, *X. fragariae*, *X. oryzae* pv. *oryzae* (Asian genetic lineage), *X. translucens* and *X. vasicola* (pvs. *musacearum* and *vasicola*). To date, all CRISPR-Cas systems found in *Xanthomonas* belong to either class I-C or I-F [275], and both clade-1 (e.g., *X. albilineans, X. hyacinthi, X. theicola*) and clade-2 strains (‘*X. badri*’, *X. cucurbitae*) were found to contain both systems [276].

*X. fragariae* was the first xanthomonad for which an evolutionary scenario of the CRISPR locus was established [192]. CRISPR spacer typing and MLVA displayed a congruent population structure, in which two major groups and a total of four subgroups of *X. fragariae* were revealed. Both CRISPR and MLVA data suggested that the two main groups were genetically separated before the first *X. fragariae* isolate was described and were potentially responsible for the worldwide expansion of the bacterial disease. Similarly, CRISPR typing of *X. citri* pv. *citri* revealed that all strains encode a CRISPR array that is built from a subset of 23 unique spacer sequences [237]. It was concluded that CRISPR typing, perhaps in combination with a minisatellite scheme, is ideally suited for placing strains associated with new outbreaks in the global diversity of *X. citri* pv. *citri*.

For both pathogens, highly informative spacers were identified based on the set of analyzed strains that allow for group determination of novel isolates for *X. fragariae* and prediction of pathotypes for *X. citri* pv. *citri* [192,237]. The presence of CRISPR spacers can be elucidated by DNA sequencing of PCR amplicons, by diagnostic PCR involving spacer-matching oligonucleotide primers or directly by oligonucleotide hybridization or using CRISPR/Cas-based detection technology [277]. However, to date, none of these technologies has been commercialized to type xanthomonads.

## 5. Genomics-Informed Approaches for Molecular Diagnostics

Within detection, identification and diagnosis, there are several realms where the ever-increasing wealth of genome sequence data has a major contribution to make. Genomics-informed methods exploit macromolecular sequences of the target pathogen and those of related non-target bacteria. For example, central to hybridization-based methods (PCR, IA methods such as LAMP [278,279,280,281], lab on a chip [282], etc.) are synthetic oligonucleotide primers whose sequences are designed to match genomic sequences that are unique to the target pathogen or pathogens. Consequently, these detect only the pathogens for which they were specifically designed. An alternative approach is the isolation of candidate causal agents from the infected material; identification and characterisation of the bacterial isolate can be greatly facilitated by genome sequencing and comparison with previously sequenced genomes. More open-ended still is shotgun metagenomic and amplicon sequencing, which allows for the unbiased discovery of even unculturable pathogens in a biological sample. Similarly, meta-transcriptomics can be useful, especially for the discovery of viral pathogens with RNA genomes, but not directly relevant for bacterial pathogens such as *Xanthomonas* species.

Analysis of genome sequences has informed the development of assays for several *Xanthomonas* pathogens (regulated and non-regulated) based on PCR [80,86,267,283,284,285] or LAMP [63,109,113,123,134,286,287]. Genomics-informed assays have not always been entirely successful. For example, a multiplex PCR [288] assay and LAMP assay [287] for the detection of the banana pathogen *X. vasicola* pv. *musacearum* targeted the *gspD* gene; unfortunately, that gene is also conserved in closely related bacteria such as *X. vasicola* pv. *vasculorum* that are not pathogenic on banana. The consequent potential for false positives required the development of new primers also informed by genomic sequence comparisons [267].

Generally, each study has used its bespoke bioinformatics workflow to identify genomic regions or protein-coding genes that are unique to the target pathogen and has then performed validation and/or assessment of assay sensitivity in vitro. Typically, the proposed primers are first tested in silico by BLASTN searches against public sequence databases or specialized virtual-amplification simulators such as Electric-LAMP [289]. Assay-development studies have employed existing tools such as Mauve [290] and BLASTN for identifying ‘islands’ of genome-specific sequence and Microscope [291] and EDGAR [292] for identifying genome-specific genes. In some cases, the method is not fully described, including steps implemented by, for example, “a custom Perl script”. However, several bioinformatics pipelines have been specifically designed for identifying discriminatory genomic targets for PCR and IA assays [293,294,295,296]. The SkIf_with_DSK tool [297] identifies specific *k*-mers within a group of genomes (in-group) that are absent in the other genome sequences (out-group), provides their precise locations on a reference genome and uses the positions to concatenate the overlapping *k*-mers into long-mers [298] on which primers can be designed for specific identification tools, as done on *X. fastidiosa* [299]. Pathogen-specific genomic sequences can be used for designing specific primers. Primer design is typically aided by software such as Primer3 and Primer Express^®^ for PCR and PrimerExplorer (https://primerexplorer.jp/e/) and LAVA [300] for LAMP.

For discrimination between closely related genotypes, it is not always possible to identify presence-absence polymorphism that serves as a suitable amplification target. Another approach is to use PCR to assay SNPs. For example, primers have been developed that discriminate between sub-clades of *X. vasicola* pv. *musacearum* based on the restriction-digestion of an amplicon that includes a single-nucleotide polymorphism [301]. This RFLP approach was informed by the comparison of complete genome sequences of representatives of each sub-clade [302]. Another discriminatory approach is the sequencing of informative loci, i.e., an MLSA; that could allow high-resolution identification of the pathogen type directly from the DNA of infected plant material, as exemplified in *Xylella* by Faino et al. [303].

A different application of genome sequencing is the rapid identification and characterisation of isolated pathogens. When a new disease emerged on beans in the Rwanda district of Nyagatare in 2013, the causative agent was isolated and displayed phenotypic features typical of *Xanthomonas,* but disease symptoms were distinct from those of known *Xanthomonas* pathogens of common beans. Genome sequencing and analysis of average nucleotide identity (ANI) with type and pathotype strains confirmed that this pathogen was not closely related to those previously known bean pathogens [304] but rather belonged to an unnamed species-level clade that was subsequently named *X. cannabis* [305]. Similarly, isolation and genome sequence analysis and ANI led to the discovery for example of two new *Xanthomonas* species associated with watercress [10] and a new species on walnut *X euroxanthea* [11].

## 6. Conclusions

Molecular methods caused a significant shift in the approaches to the detection, identification and diversity studies of plant pathogenic xanthomonads and overall have led to the development of more reliable disease management strategies. Detection and identification of a *Xanthomonas* sp. pathogen, frequently involves, as for other plant pathogenic bacteria, the isolation and culturing of the bacterium as also indicated in the diagnostic protocols. For some bacterium/plant combinations, this step from diseased plants is relatively straightforward as the abundance of bacteria on the edge of lesions associated with typical symptoms ensures that isolation plates contain many colonies with the predominance of the target pathogen. However, when analysing seeds, tree hosts, or other difficult (e.g., strawberries) or asymptomatic plant materials, the pathogenic bacteria might be in very low number. The analytical specificity and analytical sensitivity of the molecular methods might therefore allow bypassing the time-consuming and complicated isolation/culture process. In addition, suspected microorganisms may sometimes require specific biosafety requirements. Bacterial extract concentration by centrifugation, semi-selective media, immunocapture or bacterial enrichment in some cases could further improve the success of diagnosis. Another advantage of molecular techniques for diagnosis is the possibility to specifically detect and identify genetically distinct pathogens that cause similar diseases (e.g., *Xanthomonas* spp. causing tomato bacterial spot; *X. citri* pvs.) or multiple pathogens within a unique sample (as for the certification of plant material or seeds) and not less important the possibility to quantify bacterial titre by qPCR.

Most molecular methods do not inform on the viability of the bacterial inoculum. It is therefore possible that a sample tests positive, but the bacteria are no longer viable. This assumption often generates a wide debate if these plants or seeds pose a risk or not. However, as discussed in Section 3.3, protocols that assess only viable cells can be used and more research should be devoted to this area.

Cumbersome biochemical tests for *Xanthomonas* identification are nowadays outdated and have generally been replaced by amplification and sequencing of a single housekeeping gene or in combination with other genes as part of MLSA schemes. Major attention should therefore be paid to public databases. The existing metadata associated with these datasets become extremely important as a misidentified organism can affect the outcome. A possible solution is to restrict attention to data from type strains of species and pathovars. In addition, the existence of several MLSA schemes, based on different partial gene sequences, may complicate the analysis as the success of this approach is dependent on the existence of well developed, and complete, databases as discussed in Section 4.1.2.

However, there are still limits for the detection/identification of some xanthomonads characterized by high parasitic specialization. Pathovar, races or pathotypes of *Xanthomonas* spp. in some cases still need to be tested on natural or experimental plant differentials. For the most frequently found pathovars, molecular tests have been developed, but in certain cases, pathogenic and non-pathogenic strains have been found closely associated (e.g., [11,31,34]). The development of specific tests for races of a *Xanthomonas* pathogen might be possible if a race includes isolates with conserved regions, but in some cases, the races might not be monophyletic and include phylogenetically disparate isolates. In this case, the assays might need to target different groups of isolates. Ultimately, the races are determined by the phenotypes in particular host lines and race structures can be expanded as new variants are found.

Genome informed diagnostic protocols have been developed in recent years that could be further improved (e.g., [80,109,283]). Whole-genome sequences and pangenomic analyses of multiple strains belonging to all relevant xanthomonads could also facilitate the identification of new markers for the identification of different pathogens, at the species level but also markers of taxa below species and subspecies level (pathovars and races). Emerging high-throughput detection and quantification systems for pathogenic bacteria are already under investigation as they are characterised by speed, sensitivity, and ease of use. In the near future new methods of multiplexing, the detection of more markers and methods based on sequencing with a nanopore-based technology need to be developed for routine detection.

The higher speed and output-to-cost ratios of whole-genome sequencing also render this technique more and more attractive for the diagnostics of plant pathogenic bacteria although some challenges remain for a routine implementation [306]. An issue for diagnostic and diversity studies would be to also look besides plant material [6]. The study of xanthomonads in association with plants and different non-plant environments will allow for a better understanding of evolutionary processes. The increasing number of HTS studies highlight the presence of xanthomonads in unexpected samples.

An increasing number of portable molecular detection systems studied for in-field diagnosis have been developed. However, another challenge will be the development of adequate sample preparation systems that together with easy-to-use systems could result in accurate methods for early detection and surveillance that allow for timely management of plant diseases and tools for the prevention of introduction and spread of dangerous xanthomonads.

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
