# Peer review of "Trends in Molecular Diagnosis and Diversity Studies for Phytosanitary Regulated Xanthomonas"

_microorganisms, 2021, doi:10.3390/microorganisms9040862_

Round 1

Reviewer 1 Report

In my opinion, this manuscript is interesting and well written both in terms of content and language. There is always something to be done better, however, this work is thoughtful and well written. For this reason, I have no comments or queries regarding this manuscript. Congratulations to the authors.

Author Response

Dear reviewer,
We thank you for your generous evaluation, which we receive with great honour and humility.

Best regards

Reviewer 2 Report

This is a timely and comprehensive review on detection and diagnosis methods for xanthomonads. The paper is well written and organized, it is easy to read, and contains a wealth of very useful information; it will surely be a highly cited paper. In all, I think it is a very nice and valuable piece of work.

I only have a few minor comments:

  • Page 3. Last sentence is unfinished.
  • Page 5, third paragraph “a plant with symptoms [59]. Whereas ‘detection’ is..”. Authors might consider changing to something like “Conversely, ‘detection’ is…”.
  • Page 6, first sentence of heading 3.2 (“in plant bacterial diseases diagnosis those caused by xanthomonads being no exception.”) is missing something.
  • Page 8, line 3. “A few protocols on PCR or qPCR…”, could be changed to something like “A few PCR or qPCR protocols…”.
  • 9, 3rd paragraph, end of first line. I think it should be “streamlined”.
  • 10. Please, define vqPCR.
  • 11, last sentence should be modified. “the profiles of repetitive element based  PCRs  (rep-PCR…” Most of the times, the rep-PCR protocols will produce amplification patterns even if the target organism does not contain the corresponding repeats (e.g, ERIC, BOX, REP); in particular, it is doubtful that the enterobacterial ERIC repeats, for instance, are present outside the enterobacteria. Therefore, it is generally agreed that these protocol do not amplify repetitive sequences, and should be considered as arbitrarily-primed PCR protocols (see, for instance, Gillings & Holey 1997 Lett Appl Microbiol 25, 17–21 https://doi.org/10.1046/j.1472-765X.1997.00162.x).
  • Supplementary Table S1. Scientific names for plants should be in italics. Please, homogenize the name for Wilbrink's medium. Change complimentary for complementary. Correct “fingerpring”
  • Supplementary Table S2. Remove yellow highlighting. Correct the cite for Vancheva et al.
  • The number of pages are out of sequence after page 13

Author Response

Dear reviewer,

we thank you for your comments which we consider very relevant. We have addressed them all and the corresponding changes have been introduced in the manuscript. Also, several other clarifications were introduced to improve the manuscript and the reference section was updated.

In detail:

Page 3. Last sentence is unfinished.

Authors: the sentence was cut apart by the table introduction. It is now complete.

Page 5, third paragraph “a plant with symptoms [59]. Whereas ‘detection’ is..”. Authors might consider changing to something like “Conversely, ‘detection’ is…”.

Authors: the suggestion was taken in account and the sentence changed accordingly.

Page 6, first sentence of heading 3.2 (“in plant bacterial diseases diagnosis those caused by xanthomonads being no exception.”) is missing something.

Authors: the sentence was clarified

Page 8, line 3. “A few protocols on PCR or qPCR…”, could be changed to something like “A few PCR or qPCR protocols…”.

Authors: the sentence was changed has suggested.

9, 3rd paragraph, end of first line. I think it should be “streamlined”.

Authors: the correction was made.

  1. Please, define vqPCR.

Authors: the comment was addressed

11, last sentence should be modified. “the profiles of repetitive element based PCRs  (rep-PCR…” Most of the times, the rep-PCR protocols will produce amplification patterns even if the target organism does not contain the corresponding repeats (e.g, ERIC, BOX, REP); in particular, it is doubtful that the enterobacterial ERIC repeats, for instance, are present outside the enterobacteria. Therefore, it is generally agreed that these protocol do not amplify repetitive sequences, and should be considered as arbitrarily-primed PCR protocols (see, for instance, Gillings & Holey 1997 Lett Appl Microbiol 25, 17–21 https://doi.org/10.1046/j.1472-765X.1997.00162.x).

Authors agree with the reviewer comment and the sentence was changed and the reference introduced in the manuscript.

Supplementary Table S1. Scientific names for plants should be in italics. Please, homogenize the name for Wilbrink's medium. Change complimentary for complementary. Correct “fingerpring”

Authors: the suggested corrections were introduced in Table S1. Also, several references were added to the reference list since they were missing in the former version.

Supplementary Table S2. Remove yellow highlighting. Correct the cite for Vancheva et al.

Authors: Table S2 was corrected, and references numbers were updated.

The number of pages are out of sequence after page 13

Authors: pages were updated has requested.

Best regards